# Hydration and Mobility of Alkaline Metal Cations in Sulfonic Cation Exchange Membranes

**DOI:** 10.3390/membranes13050518

**Published:** 2023-05-16

**Authors:** Vitaly I. Volkov, Nikita A. Slesarenko, Alexander V. Chernyak, Irina A. Avilova, Victor P. Tarasov

**Affiliations:** 1Federal Research Center of Problems of Chemical Physics and Medicinal Chemistry RAS, 142432 Chernogolovka, Russia; wownik007@mail.ru (N.A.S.); sasha_cherniak@mail.ru (A.V.C.); irkaavka@gmail.com (I.A.A.); tarasov.07@list.ru (V.P.T.); 2Scientific Center in Chernogolovka of the Institute of Solid State Physics Named Yu. A. Osipyan RAS, 142432 Chernogolovka, Russia

**Keywords:** sulfonic cation-exchange membrane, hydration number, sulfonated polystyrene, pulsed field gradient NMR, NMR relaxation, diffusion coefficient, ionic conductivity, correlation time, mass transfer, selectivity mechanism

## Abstract

The interconnection of ionogenic channel structure, cation hydration, water and ionic translational mobility was revealed in Nafion and MSC membranes based on polyethylene and grafted sulfonated polystyrene. A local mobility of Li^+^, Na^+^ and Cs^+^ cations and water molecules was estimated via the ^1^H, ^7^Li, ^23^Na and ^133^Cs spin relaxation technique. The calculated cation and water molecule self-diffusion coefficients were compared with experimental values measured using pulsed field gradient NMR. It was shown that macroscopic mass transfer is controlled by molecule and ion motion near sulfonate groups. Lithium and sodium cations whose hydrated energy is higher than water hydrogen bond energy move together with water molecules. Cesium cations in possession of low hydrated energy are directly jumping between neighboring sulfonate groups. Cation Li^+^, Na^+^ and Cs^+^ hydration numbers (*h*) in membranes were calculated from ^1^H chemical shift water molecule temperature dependences. The values calculated from the Nernst–Einstein equation and the experimental conductivity values were close to each other in Nafion membranes. In MSC membranes, calculated conductivities were one order of magnitude more compared to the experimental ones, which is explained by the heterogeneity of the membrane pore and channel system.

## 1. Introduction

Understanding of ion exchange membrane selectivity nature is a fundamental problem of ionic separation [1,2]. Ionic transport in sulfonic cation (-exchange) membranes is controlled by the ionogenic channel nanostructure and particularities of the cation hydration. For the transport mechanism revelation of an ionic interaction, the mobility of water molecules and cations in different spatial scales should be investigated. NMR spectroscopy is widely applied to the structure and dynamics of complicated molecular systems studies. Nuclear spin relaxation and pulsed field gradient NMR (PFG NMR) became the methods of choice for molecular and ionic mobility revelation in polymeric electrolytes. The more interesting and fundamental results were obtained for water molecule and alkaline cations transport in sulfocation-exchange membranes. Perfluorinated Nafion [3,4,5,6,7,8,9,10,11,12,13,14,15,16,17,18,19,20] and MF-4SC membranes [21,22,23,24] attracted a special interest. Self-diffusion of water molecules [3,4,5,6,7,8,9,10,11,12,13,14,15,16,17,18,19,21,22,23,24] and Li^+^, Na^+^ and Cs^+^ cations were investigated at different water contents using PFG NMR [4,13,20,21]. The correlation between cation hydration numbers (*h*) and self-diffusion coefficients was revealed. Spin relaxation investigations of ^1^H water molecule and cation nuclei gave us the opportunity to calculate water and Li^+^, Na^+^ and Cs^+^ translation correlation times in appropriate ionic forms in Nafion and MF-4SC membranes. It was concluded that the local mobility of molecules and cations near sulfonate groups governs its macroscopic transport [20,21]. The mobility of alkaline cations has been investigated rather poorly in spite of some publications devoted to the relaxation and diffusion of ^7^Li, ^23^Na and ^133^Cs in Nafion and polystyrene sulfonate systems [20,21,25,26,27,28,29,30] showing the efficiency of a combined application of spin relaxation and PFG NMR techniques. Local cation mobilities research into lithium and the sodium salts of polystyrene sulfonic acid is especially interesting [27,28,29,30]. These investigations are fundamental because polystyrene sulfonic acid and salt aqueous solutions may be treated as models of homogeneous membranes—for instance, of MSC membranes based on polyethylene and grafted sulfonated polystyrene. These studies are especially interesting for cations with different hydration abilities, such as Li^+^, Na^+^ and Cs^+^; therefore, they enable us to understand microscopic transport mechanism.

In the last decade, the number of works devoted to ionic conductivity and the self-diffusion of water molecules, hydrated H^+^ cations and alkaline metal ions Li^+^, Na^+^ and Cs^+^ in Nafion membranes and membranes based on polyethylene with grafted sulfonated polystyrene (MSC) has increased [25,26,31]. Comparison of PFG NMR and impedance spectroscopy measurements gave us the opportunity to reveal the main features of the translational mobility of water molecules, alkaline metal cations and ionic conductivity in sulfocation-exchange MSC and Nafion 117 membranes. The ionic conductivities of Li^+^, Na^+^ and Cs^+^ in Nafion and MSC membranes are changed in the same way as the cation self-diffusion coefficients. The dependences of ionic conductivities and self-diffusion coefficients on temperature and moisture content are symbatic. It was shown that it becomes possible to purposefully vary the mobility of cations and, consequently, the selectivity of membranes to cations of alkali metals. This may be done, for example, by changing the moisture content of MSC through spatial crosslinking of its polymer chains.

The mobility and self-diffusion peculiarities of water and Li^+^, Na^+^ and Cs^+^ cations in the Nafion membrane and in the MSC membrane were revealed by the ^1^H, ^7^Li, ^23^Na and ^133^Cs spin relaxation and PFG NMR techniques discussed. These membranes are an example of the main type of sulfo cation exchangers: perfluorinated and sulfonated polystyrene membranes.

## 2. Materials and Methods

### 2.1. Materials

Extruded N117 (thickness 183 μm, equivalent weight (EW) = 1100, Dupont, Ion Power Inc., New Castle, DE, USA) membranes were used for the experimental characterization of Nafion in salt (Li^+^, Na^+^, Cs^+^) ionic form. As-received membranes were pre-cleaned by boiling in 3 wt % oxygen peroxide (H_2_O_2_) for 1 h to eliminate organic residues and rinsed with demineralized water. To guarantee a complete acidification and paramagnetic ion impurities extraction, the samples were soaked in aqueous 10 M H_2_SO_4_ at room temperature for 1 h and rinsed with demineralized water. Finally, the membranes were boiled for 1 h in aqueous 1 M HCl and then rinsed with demineralized water. Salt membrane ionic forms were prepared through repeated equilibration with aqueous lithium, sodium and cesium chloride 1M solutions, followed by rinsing in demineralized water [4,5,13]. For the required humidity achievement, Nafion samples were placed in an open weighing bottle inserted in desiccators in which the relative humidity was determined by the vapor pressure of the saturated aqueous salt solution [5,6,13]. Membrane samples in contact with water were also investigated. Sulfocation-exchange MSC membranes made from polyethylene with sulfonated grafted polystyrene were studied. These membranes were obtained through postradiation graft polymerization of styrene on a low-density polyethylene film previously peroxidized through irradiation in air on a ^60^Co γ-radiation source, followed by sulfonation of the grafted polystyrene with 96% sulfuric acid. The concentration of the sulfo groups, the static exchange capacity (SEC), was 2.5 meq/g. The membrane thickness was more than 20 µm. A detailed method for obtaining MSC membranes is described in our previous works [25,26,31].

For NMR measurements, membrane plates with a dimension of 3 mm × 40 mm were inserted in hermetically closed NMR sample tubes, the outer diameter of which was 5 mm. Dry membrane samples were obtained by drying at 110 °C until a constant weight or equilibration with P_2_O_5_ at room temperature. The gravimetrical measured water content *λ_g_* was characterized as water molecule amount per sulfonated group, calculated from Equation (1):(1)$$\lambda_{g}=\frac{m_{H_{2}O}\cdot EW}{m_{dry}\cdot M(H_{2}O)},\;\mathrm{or}\;\lambda_{g}=\frac{m_{H_{2}O}}{m_{dry}\cdot M(H_{2}O)\cdot IEC}$$
where *EW* is the equivalent weight and *IEC* is the ion-exchange capacity of the membrane considered. In this study, we used the values of *EW* = 1100 g/eq and *IEC*= 2.5 mg-eq/g. *M* (*H*_2_*O*) = 18 g/mol is the molar mass of water, *m_H_*_2*O*_ is the total mass of water in the membrane and *m_dry_* is the mass of the dry membrane.

### 2.2. High-Resolution NMR

High resolutions of ^1^H, ^7^Li, ^23^Na and ^133^Cs membrane spectra were recorded on the AVANCE-III-500 Bruker NMR spectrometer (the proton Larmor frequency was 500 MHz) in the temperature region from −40 °C to +60 °C. The ^1^H chemical shift was calculated relative to the bulk water ^1^H NMR signal (*δ*_H2O_ = 4.30 ppm relatively TMS), and the chemical shift measurement error was less than 0.05 ppm.

NMR spectra of ^1^H, ^7^Li, ^23^Na and ^133^Cs nuclei belong to water molecules, and Li^+^, Na^+^ and Cs^+^ cations are singlet lines whose width is rather narrow even at temperatures below 0 °C. A high water and cation mobility is indicated at low temperatures. An example of the temperature evolution of water molecule ^1^H spectra in Cs^+^ Nafion ionic form is shown inserted in Figure 1.

### 2.3. PFG NMR

The self-diffusion coefficients were measured for ^1^H, ^7^Li, ^23^Na and ^133^Cs nuclei with the pulsed field gradient technique at the frequencies 400.22, 155.51, 105.84 and 52.48 MHz, respectively. The measurements were carried out on a Bruker AVANCE-III-400 NMR spectrometer equipped with the diff60 gradient unit. The pulsed field gradient stimulated echo sequence was used. Three 90° pulses produced a stimulated spin echo at time 2*τ* + *τ*_1_ (where *τ* is the time interval between the first and second 90° pulses and *τ*_1_ is the interval between the second and the third pulses). The magnetic field gradient pulses of amplitude *g* and duration *δ* were applied after the first and third 90° pulses. The gradient strength was varied linearly in 32 steps within a range from 0.1 to 27 T/m in value. The integrated intensities of spectrum lines were used to obtain the dependence of echo signal attenuation on *g*^2^ (diffusion decay) [19,20].

The evolution of the spin echo signal is described by the following equation.
(2)$$A(2\tau ,\tau_{1},g)=A(2\tau ,\tau_{1})\exp(-\gamma^{2}g^{2}\delta^{2}t_{d}D_{s})$$
where *γ* is the gyromagnetic ratio, Δ is the interval between gradient pulses, *t_d_* = Δ − *δ*/3 is the diffusion time and *D_s_* is the self-diffusion coefficient. *A*(2*τ*, *τ*_1_, 0) is expressed by the equation
(3)$$A(2\tau ,\tau_{1},g)=\frac{A(0)}{2\exp\left(-\frac{2\tau }{T_{2}}-\frac{\tau_{1}}{T_{1}}\right)}$$
where *A*(0) is the signal intensity after the first radio frequency (RF) pulse. *T*_1_ and *T*_2_ are the spin–lattice and spin–spin relaxation times, respectively. During the measurement of echo signal evolution, *τ* and *τ*_1_ were fixed, and only the dependence of *A* on *g* (diffusion decay) was analyzed.

Experimental diffusion decays were well approximated by Equation (2) in 2–3 orders of magnitude; self-diffusion coefficient measurement error was less than 10%.

### 2.4. NMR Relaxation

Spin–lattice *T*_1_ and spin–spin *T*_2_ nuclear relaxation times were measured using 180° -*τ*-90° and Carr–Purcell–Meiboom–Gill (90° -*τ*-*n*180°) pulsed sequences, correspondingly. Longitudinal magnetization *M_z_* recovery and perpendicular magnetization *M_x·y_* decay were approximated using exponential dependences (3) and (4), correspondingly, for ^1^H, ^7^Li, ^23^Na and ^133^Cs nuclei. The spin echo attenuation of ^23^Na and ^133^Cs nuclei was too fast to detect a detailed decay curve shape.
(4)$$\frac{\left(M_{0}-M_{z}\right)}{2M_{0}}=\exp(-t/T_{1})]$$

*M_o_* is equilibrium nuclear magnetization.

## 3. Results and Discussion

### 3.1. Hydration Numbers Calculation

From water molecules ^1^H chemical shift temperature dependences (Figure 1), the hydration numbers *h* of Li^+^, Na^+^ and Cs^+^ cations in Nafion 117 and MSC membranes were calculated (Table 1) using Equation (5) [4,5,13,26].
(5)$$h=\lambda_{s}\left[1-\frac{\frac{d\delta }{dT}}{\frac{d\delta_{H_{2}O}}{dt}}\right]$$
where *dδ*/*dt* is a chemical shift temperature dependence of membrane water, *dδ_H_*_2*O*_/*dt* is the chemical shift temperature dependence of bulk water and *λ_s_* is the number of water molecules per SO_3_^−^ group. In some papers, [32] for example, *λ_s_* is wrongly called “hydration number”.

The *h* values are 4.1 ± 1, 5 ± 1 and 3.1 ± 1 and 5 ± 1, 6 ± 1 and 1 ± 0.2 for Li^+^, Na^+^ and Cs^+^ cations in MSC [26] and Nafion membranes [4], correspondingly. The latter values are closed to *h* of these cations in aqueous chloride solutions [13]. At high water content (RH = 98%) in MSC membranes, cation self-diffusion coefficients increased in the row Li^+^ < Na^+^ < Cs^+^ [26]. The cation self-diffusion coefficients sequence was Li^+^ ≥ Na^+^ > Cs^+^ in Nafion membranes [4]. The discrepancy of the cation translation mobility row in these membranes might be explained by Cs^+^—SO_3_^−^ group contact ionic pairs forming in Nafion as opposed to in MSC. 

### 3.2. ^1^H Relaxation, Local Water Mobility and Self-Diffusion

The temperature dependences of spin–lattice (*T*_1_) and spin–spin (*T*_2_) relaxation times are shown in Figure 2.

The spin of ^1^H nuclei is 1/2; therefore, spin relaxation occurs due to proton dipole–dipole interaction modulated by water molecule mobility. In the case of the exponential correlation function relaxation times described by Bloembergen, Purcell and Pound (BPP) equations:(6)$$\frac{1}{T_{1}}=\frac{2}{3}\gamma^{2}\langle \Delta H^{2}\rangle \left(\frac{\tau_{c}}{1+{\left(\omega \tau_{c}\right)}^{2}}+\frac{4\tau_{c}}{1+{\left(\omega \tau_{c}\right)}^{2}}\right)$$
(7)$$\frac{1}{T_{2}}=\frac{1}{3}\gamma^{2}\langle \Delta H^{2}\rangle \left(3\tau_{c}+\frac{5\tau_{c}}{1+{\left(\omega \tau_{c}\right)}^{2}}+\frac{2\tau_{c}}{1+{\left(2\omega \tau_{c}\right)}^{2}}\right)$$
(8)$$\langle \Delta H^{2}\rangle =\frac{9\gamma^{2}\hbar^{2}}{20r^{6}}$$
(9)$$\tau_{c}=\tau_{c_{0}}exp\left[E_{c}/RT\right]$$
where *γ* is proton gyromagnetic ratio, *r* is proton–proton distance, *ω* is proton NMR frequency and *τ_c_* is correlation time. In this case, the dependence of *T*_1_ on 1*/T* is Lorentz shaped and shows at a minimum condition of *ωτ_c_* = 0.62. At this temperature, *T*_1_*/T*_2_ is equal to 1.6. As is shown in Figure 2, *T*_1_ (1/*T*) dependences are not approximated BPP equations and *T*_1_/*T*_2_ > 1.6. It is indicated on the distribution of correlation time. The shapes of the distribution function of water proton relaxation in perfluorinated sulfocation-exchange membranes were analyzed [24]. Two types of distribution function, Gaussian and rectangular, were applied. In spite of the correlation times distribution, the condition of *T*_1_ minimum was the same at *ωτ_av_* = 0.62, where *τ_av_* is the average correlation time [19,23,24]. We applied a Gaussian distribution (Figure 2). At these temperatures, correlation times at minimum *T*_1_(*T*) were 2.46 × 10^−10^ s and 1.97 × 10^−10^ s for resonance frequencies 500 MHz and 400 MHz, correspondingly. The water self-diffusion coefficient may be estimated from the Einstein relation [19]:*D* = *l*^2^/6 *τ_av_*(10)
where *l* is average jumping distance. Calculated self-diffusion coefficients were compared with experimental values at temperatures equal to a temperature of minimum *T*_1_, which is indicated in Table 2. We may estimate the diffusion length of cation l in the PFG NMR experiment as (6Dt_d_)^1/2^. The minimum t_d_ (diffusion time) is 10^−2^ s and D (self-diffusion coefficient) is 10^−11^ m^2^/s [5], so l is about 1 µm. It is macroscopic size. Therefore, in PFG NMR, we measure the macroscopic self-diffusion coefficient. The correlation times may be calculated correctly at temperatures where the spin relaxation time *T*_1_ shows a minimum. We measured self-diffusion coefficients namely at these temperatures in order to compare calculated and experimental self-diffusion coefficients.

The distance *l* was calculated in Nafion 117 for different humidities. For Li+ and Na+, the ionic forms’ water jumping distances were 0.15–0.2 nm at low water content and about 3.0 nm at high humidity (Table 2). The first values were comparable with water hydrogen bond length, but the second was equal to water molecule size. Therefore, it may be concluded that macroscopic water self-diffusion is controlled by water molecule local motion through the continuous hydrogen bond network which forms at rather high water content (*λ* is 6.4 in Li^+^ form and 5.1 in Na^+^ form at RH = 86%). 

### 3.3. ^7^Li, ^23^Na and ^133^Cs Spin Relaxation, Local Cation Mobilities and Self-Diffusion

The nuclear spins of ^7^Li and ^23^Na are 3/2, but the ^133^Cs nuclear spin is 7/2. For these nuclei, the main relaxation mechanism is quadrupole relaxation. For nuclear spin 3/2
(11)$$\frac{\left(M_{z}-M_{0}\right)}{M(\cos\Theta -1)}=\frac{1}{5}\exp(-2J_{1}t)+\frac{4}{5}\exp(-2J_{2}t)$$
(12)$$\frac{\left(M_{x}\right)}{M_{0}}\cdot \sin\Theta =\frac{3}{5}\exp[-(J_{0}+J_{1})t]+\frac{2}{5}\exp[-(J_{1}+J_{2})t]$$
where *ω* is NMR frequency, Ѳ is a rotation angle of equilibrium magnetization and *M*_0_ during radio frequency pulse *J*(*λω*) is the spectral density on the frequency *λω* (*λ* = 0, 1, 2).
(13)$$J_{\lambda }=0,1\cdot \pi^{2}\cdot \chi^{2}\cdot J(\lambda \omega )$$
where
*χ = eQ·eq/h*(14)

*Q* is the nuclear quadrupole moment, *eq* is the mean square value of the electric field gradient on the nucleus and *h* is Planck’s constant.
(15)$$J(\lambda \omega )=\frac{2\tau }{\left[1+{\left(\lambda \omega \tau \right)}^{2}\right]}$$

*τ = τ_o_·exp(E_a_/RT)*, where *τ* is the correlation time and *E_a_* is the activation energy [20,21].

Spin–lattice and spin–spin ^7^Li, ^23^Na and ^133^Cs nuclei relaxation times temperature dependences in Nafion 117 membranes are shown in Figure 3.

The *T*_1_(*T*) dependences show at a minimum of (*ωτ*)^2^ ≈ 1. Cation self-diffusion coefficients calculated from Equation (10) were compared with experimental ones.

As is shown in Table 3, for Li^+^ cations the jumping distance *l* is 0.15–0.18 nm, which is close to water hydrogen bond length. It may be supposed that Li^+^ cation translation displacement is controlled by a rearrangement of hydration water molecule hydrogen bonds that explains a symbasis of water and lithium cation diffusion behavior. For Cs^+^ cations, *l* is approximately 0.7 nm. This value is equal to the average distance between neighboring sulfonate groups. Therefore, it may be assumed that *τ,* estimated from ^133^Cs spin–lattice relaxation, is the time of cesium cations jumping between SO_3_^−^ groups. It looks very likely because the electric field gradient *eq* is dramatically changed when a cation comes to or departs from the sulfonate group.

Ionic conductivities were calculated on the basis of the Nernst–Einstein Equation (16):(16)$$\sigma =\frac{N\cdot D\cdot e^{2}}{k\cdot T}$$
where *N* is the number of charge carriers, cm^3^; *D* is the self-diffusion coefficient, m^2^/s; *e* is the electron charge, 1.9 × 10^–19^ C; *k* is the Boltzmann constant, 1.38 × 10^–23^ J/K; and *T* is the absolute temperature, K.

The calculated conductivities were compared with experimental values measured using impedance spectroscopy. As is shown in Table 4, cation self-diffusion coefficients calculated from local mobility *D_R_* were in good agreement with experimental self-diffusion coefficients *D_PFG_* measured from PFG NMR. Ionic conductivities calculated from Equation (16) were much closer to experimental values measured using impedance spectroscopy.

All results listed above are explained on the basis of the nanostructural model of transport channels in Nafion shown in Figure 4.

According to this model, ionic and water transport is realized through nanochannels of infinite length by jumping between neighboring sulfonate groups. The main centers of hydration are cations; water molecules also form hydrogen bonds with sulfonate group oxygen. Water molecules, cations and sulfonate groups form channels of infinite length for ionic macroscopic transfer (broken curve in Figure 4). Water content changes the channel width only [6].

In Figure 5, fragments of transport channels are shown in the Li^+^ and Cs^+^ ionic forms of Nafion 117. Nanochannels for cesium cations are narrowly compared to lithium cations because of lower humidity in Cs^+^ ionic form. In Li^+^ ionic form, water molecules form a continuous hydrogen bond network, and the lithium cation moves through low dimension steps which are hydrogen bond length. In Cs^+^ ionic form, the hydrogen bond network is broken; therefore, the cesium cation is jumping directly between the nearest SO_3_^−^ groups.

In the MSC membrane, the temperature dependences of relaxation times are similar to spin relaxation behavior in the Nafion membrane. Spin–lattice relaxation times temperature dependences *T*_1_ (*T*) also display at minimum, as is shown in Figure 6. Correlation times were calculated at the temperature of *T*_1_ minimum. The self-diffusion coefficients of cations were calculated based on the Einstein relation.

In Table 5, the self-diffusion coefficients of Li^+^, Na^+^ and Cs^+^ calculated from NMR relaxation and measured using PFG NMR compared with calculated and measured ionic conductivities in MSC membranes are shown.

The calculated and experimental values of the self-diffusion coefficients are rather close to each other. However, calculated conductivities are one order of magnitude more compared with experimental values. As was shown earlier, the distribution of water molecules in MSC membranes is uneven: some of the water molecules are located in the ionogenic phase, while the rest are localized in areas with a depleted content of sulfo groups and cations, as is shown in Figure 7. In these two areas, which are connected sequentially to each other by channels, cations are well hydrated; therefore, cation behavior in MSC membranes is similar to that in aqueous salt solutions. During macroscopic transfer, cations are moving in high and low sulfo group concentration regions. In sulfo-group-depleted parts, ionic conductivity is lower than in charge-rich parts. The depleted membrane part namely limits macroscopic conductivity.

## 4. Conclusions

Spin relaxation and pulsed field gradient NMR were applied to ^1^H, ^7^Li, ^23^Na and ^133^Cs nuclei in order to reveal water and cation transfer particularities in different spatial scales in Nafion 117 and MSC membranes. Spin–lattice and spin–spin relaxation times temperature dependences analysis gave us the opportunity to estimate the correlation times of local water molecules and Li^+^, Na^+^ and Cs^+^ cations motion. Self-diffusion coefficients calculated from the Einstein relation were compared with the values of macroscopic self-diffusion coefficients measured using pulsed field gradient NMR. It was concluded that macroscopic mass transfer in Nafion membranes is controlled by molecule and ion motion near sulfonate groups. This conclusion is explained from the point of view of the structural model of membrane ionogenic channels. Lithium and sodium cations translation displacement is controlled by a rearrangement of hydration water molecules hydrogen bonds that explains a symbasis of water and lithium cation self-diffusion behavior. It may be assumed that, opposite to Li^+^ and Na^+^, cesium cations jump between SO_3_^−^ groups directly. It looks very likely because the electric field on the cation nucleus is dramatically changed when the cation comes to or departs from the sulfonate group. In MSC membranes based on polyethylene with sulfonated grafted polystyrene, the diffusion coefficients estimated from the correlation times are also close to the diffusion coefficients measured using the NMR PFG method. Ionic conductivities calculated from cation self-diffusion coefficients are one order of magnitude more compared to experimental values. It has been shown that the limiting stage of macroscopic cation transfer ion transport is ion transport in the regions with a lower concentration of sulfo groups.

## Figures and Tables

**Figure 1 membranes-13-00518-f001:**
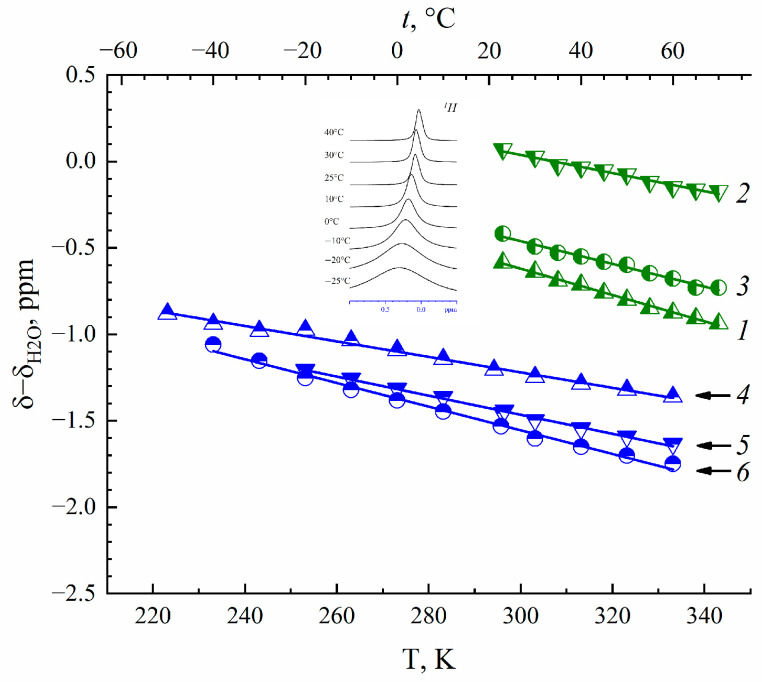
Temperature dependence of water molecule ^1^H nuclear chemical shift in Li^+^(1), Na^+^(2) and Cs^+^(3) ionic forms of MSC membrane and Li^+^(4), Na^+^(5) and Cs^+^(6) ionic forms of Nafion 117 membrane at RH = 95%. Evolution of ^1^H (water molecules) spectra with temperature variation in Cs^+^ Nafion ionic form is shown in insertion.

**Figure 2 membranes-13-00518-f002:**
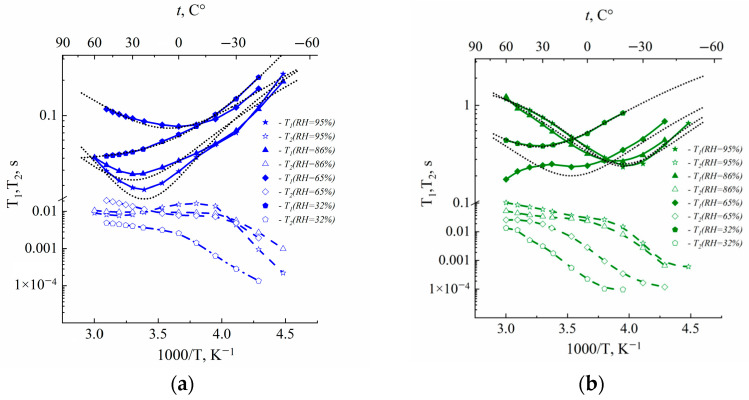
Temperature dependences of ^1^H spin–lattice *T*_1_ and spin–spin *T*_2_ relaxation times of water molecules in Li^+^ (**a**)*,* Na^+^ (**b**) and Cs^+^ (**c**) ionic forms in Nafion 117 membrane at different RH. Dotted lines are Gaussian functions.

**Figure 3 membranes-13-00518-f003:**
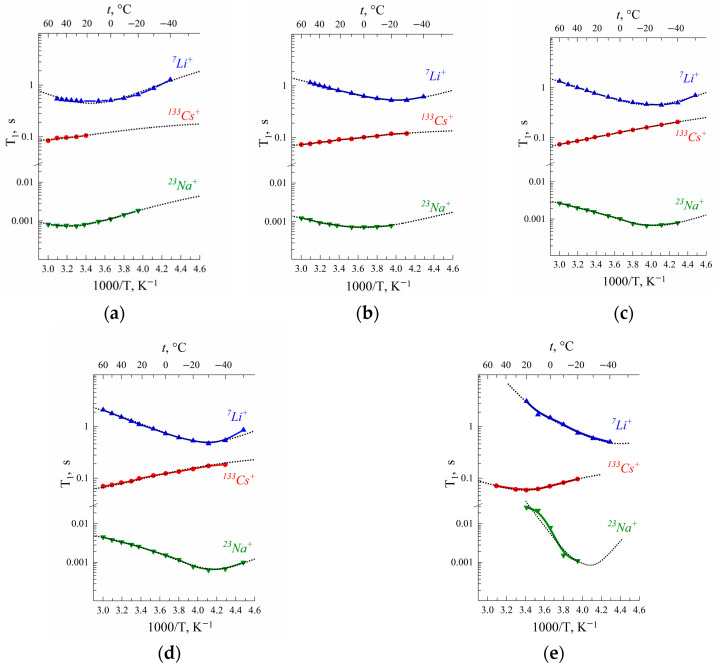
Temperature dependences of ^7^Li, ^23^Na and ^133^Cs nuclei spin–lattice *T*_1_ relaxation times in Nafion at RH (**a**) 32%, (**b**) 65%, (**c**) 86%, (**d**) 95% and (**e**) 98% [20]. Dotted lines are Gaussian function approximation.

**Figure 4 membranes-13-00518-f004:**
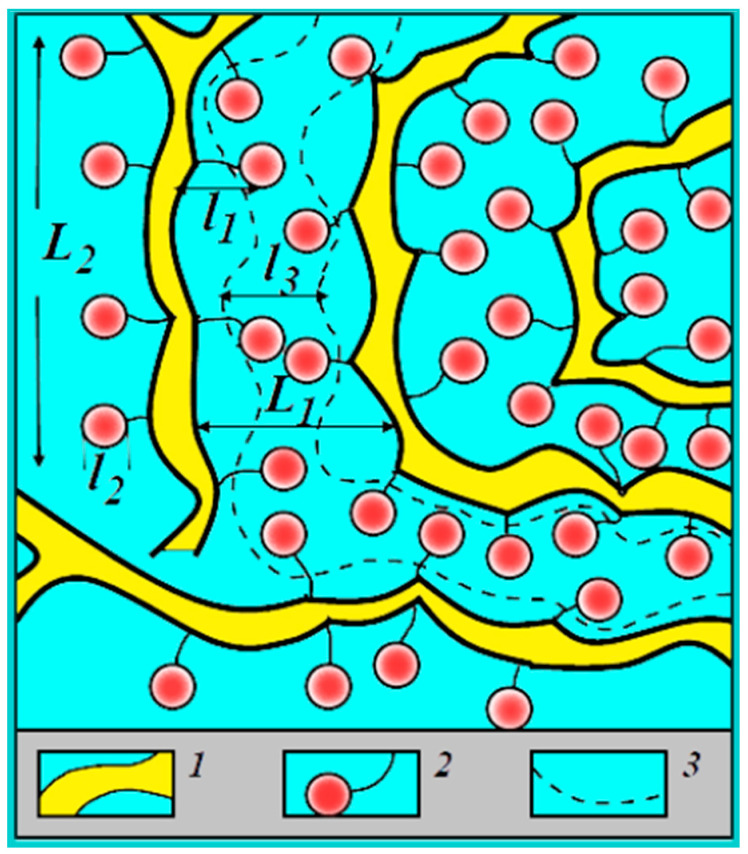
Structure of the amorphous part of a perfluorinated sulfonate cation-exchange membrane [6]. (1) Polymer backbone; (2) hydrated counter-ions and functional groups at a low moisture content; (3) transport channels for ions and water molecules at a high moisture content; L_1_ = 4 nm according to low-angle X-ray scattering data; L_2_ = 10 nm according to Mössbauer spectroscopy; l_1_ = l2 = 1 nm according to ENDOR and relaxation NMR data; l_3_ = 1.5 nm according to standard porosimetry and ENDOR methods.

**Figure 5 membranes-13-00518-f005:**
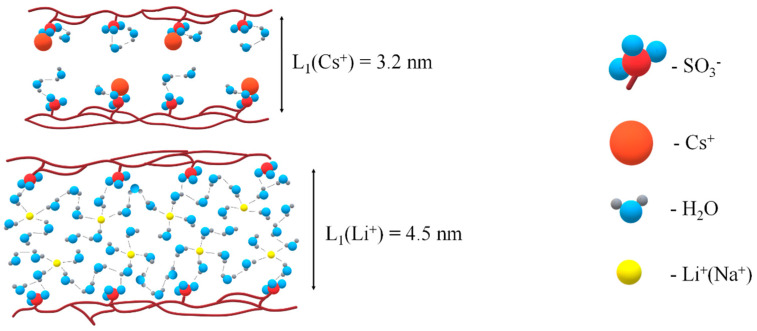
Fragments of ionogenic nanochannels in Li^+^ and Cs^+^ ionic forms of Nafion membranes.

**Figure 6 membranes-13-00518-f006:**
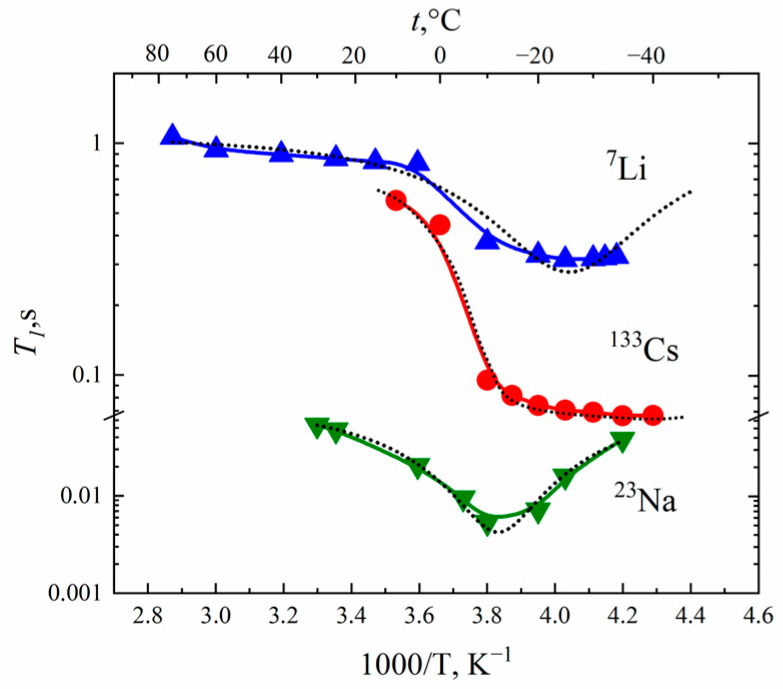
Temperature dependences of the spin–lattice relaxation times *T*_1_ of ^7^Li, ^23^Na and ^133^Cs nuclei in appropriate ionic forms of MSC membranes. Dashed curves are Lorentzian functions.

**Figure 7 membranes-13-00518-f007:**
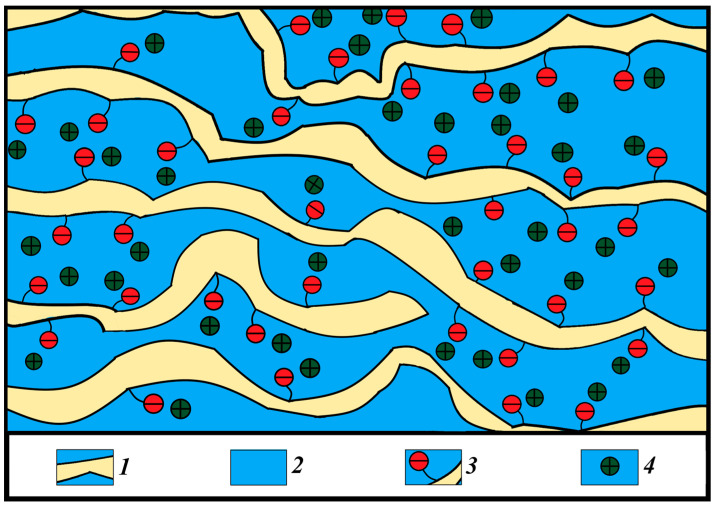
Scheme of the structure of the system of pores with different sulfo group density connected by channels in MSC membranes. (1) Polymer backbone; (2) water channels; (3) SO_3_^−^ group; (4) Li^+^, Na^+^ and Cs^+^ cations; designed on the basis of MSC nanostructure given in [25].

**Table 1 membranes-13-00518-t001:** Water content *λ_s_* and hydration number *h* of Li^+^, Na^+^ and Cs^+^ cations in Nafion and MSC membranes at RH = 98% and in equimolar aqueous salt chloride solutions.

	Nafion [4]			MSC [26]			Aqueous Chloride Solutions [13]		
Ionic Form	Li^+^	Na^+^	Cs^+^	Li^+^	Na^+^	Cs^+^	Li^+^	Na^+^	Cs^+^
*λ_s_*, [H_2_O]/[SO_3_^−^] or concentration	12	10	3	13.8	10.3	8.1	Equi- molar to MSC	Equi- molar to MSC	Equi- molar to MSC
*h*	5.0 ± 1.0	6.0 ± 1.0	1.0 ± 0.2	4.1 ± 1.0	5.0 ± 1.0	3.1 ± 1.0	4	4.6	3.9

**Table 2 membranes-13-00518-t002:** Water self-diffusion coefficients measured using PFG NMR (*D_s_*) at conditions of minimum *T*_1_ for Li^+^, Na^+^ and Cs^+^ Nafion membrane ionic forms at different RH. The variable *l* is the water molecule proton jumping length calculated from the Einstein relation.

Ionic Form	Li^+^			Na^+^			Cs^+^		
*RH*, *%*	65	86	95	65	86	95	65	86	95
*T*_1*min*_ (*T*), °C	0	30	22	10	−20	−30	40	−10	22.6
*D_s_* at *T*_1*min*_ (*T*), m^2^/s (10^−11^)	3.0	10.0	30.0	0.92	2.0	3.7	0.7	0.29	1.1
*l*, nm	0.2	0.28	0.32	0.1	0.15	0.21	0.1	0.1	0.11

**Table 3 membranes-13-00518-t003:** Li^+^ and Cs^+^ cation jumping length *l* in Nafion membranes at different humidity.

Ionic Form	Li^+^			Cs^+^
*λs*, [H_2_O]/[SO_3_^−^] [4]	5.7	8.9	10.7	4
*T*_1*min*_ (*T*), °C	−20	−30	−30	20.2
*D_s_* at *T*_1*min*_ (*T*), m^2^/s (10^−12^)	4.5	4.7	6.5	30.0
*l*, nm	0.15	0.15	0.18	0.7

**Table 4 membranes-13-00518-t004:** Self-diffusion coefficients of cations at 293 K calculated from the Einstein relation on the basis of NMR relaxation data (*D_R_*). Self-diffusion coefficients *D_PFG_* measured using PFG NMR. Ionic conductivities (*σ_exp_*) measured using impedance spectroscopy and calculated (*σ_calc_*) from the Nernst–Einstein relation on the basis of self-diffusion coefficients of Li^+^, Na^+^ and Cs^+^ cations in a Nafion membrane.

Ionic Form	*D_PFG_*, m^2^/s [4]	*D_R_*, m^2^/s	*σ_exp_*, S/m [4]	*σ_calc_*, S/m [4]
Li^+^	(1.5 ± 0.1) × 10^−10^	(1.0 ± 0.1) × 10^−10^	1.3 ± 0.1	1.6 ± 0.1
Na^+^	(2.0 ± 0.3) × 10^−10^	(0.8 ± 0.1) × 10^−10^	1.1 ± 0.1	2.0 ± 0.3
Cs^+^	(0.6 ± 0.2) × 10^−10^	(0.34 ± 0.1) × 10^−10^	(2.3 ± 0.3) × 10^−1^	(6.0 ± 0.2) × 10^−1^

**Table 5 membranes-13-00518-t005:** Self-diffusion coefficients at 293 K calculated on the basis of the Einstein equation from the NMR relaxation data (*D_R_*), experimental self-diffusion coefficients measured using pulsed magnetic field gradient NMR (*D_PFG_*), ionic conductivity measured using impedance spectroscopy (*σ_exp_*) and calculated (*σ_calc_*) on the basis of the Nernst–Einstein relation from the self-diffusion coefficients of Li^+^, Na^+^ and Cs^+^ cations in a MSC membrane [25].

Ionic Form	*D_PFG_*, m^2^/s [25]	*D_R_*, m^2^/s [25]	*σ_exp_*, S/m	*σ_ca_*_l*c*_, S/m
Li^+^	5.0 × 10^−10^	3.8 × 10^−10^	2.7 × 10^−1^	5.3
Na^+^	5.0 × 10^−10^	5.5 × 10^−10^	3.0 × 10^−1^	5.0
Cs^+^	9.0 × 10^−10^	1.5 × 10^−10^	8.0 × 10^−1^	9.0

## Data Availability

Not applicable.

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
