# Peer review of "Hydration and Mobility of Alkaline Metal Cations in Sulfonic Cation Exchange Membranes"

_membranes, 2023, doi:10.3390/membranes13050518_

Round 1
Reviewer 1 Report
The manuscript reported the influence of ionogenic channel structure, cation hydration, water and ionic translational mobility interconnection on the sulfonic CEMs. Nafion membranes and MSC based on polyethylene and grafted sulfonated polystyrene are mainly investigated. The local mobility of Li+, Na+, Cs+ cations and water molecules was estimated by 1H, 7Li, 23Na, 133Cs spin relaxation technique. The calculated cation and water molecule self-diffusion coefficients were compared with experimental values measured by pulsed field gradient NMR. It is found that the calculated conductivity from the Nernst-Einstein equation and experimental conductivity values are close to each other in Nafion membranes. In MSC membranes calculated conductivities are one order of magnitude more comparing the experimental ones which may be explained by the heterogeneity of membrane pore and channel system.
The content of this manuscript meets the reading interests of the readers of the journal. However, there are certain English spelling and grammar issues, and also the discussion and explanation should be further improved. I suggest giving a minor revision and the authors need to clarify some issues or supply some more experimental data to enrich the content.
1. For grammar issues, it is suggested that the author double-check the small grammar errors in the full text, especially the lack of and redundant use of definite articles.
2. For the Keywords, 'selectivity mechanism', 'sulfonated polystyrene', and 'mass transfer' should be added in order to attract a broader readership.
3. Obviously, the length of the abstract is too long, significantly exceeding the requirements of the journal. The content of the abstract needs to be significantly reduced. 'Abstract: The abstract should be a total of about 200 words maximum. ' (see https://www.mdpi.com/journal/membranes/instructions).
4. Page 1, 'Selectivity mechanism of ion exchange systems is a fundamental problem of membrane separation [1,2].' It should be introduced as the result of low selectivity. For example, for many electrochemical applications, if the IEM cannot act as an efficient barrier layer to the redox-active species, the crossover will result in very low efficiency (Electrochimica Acta 378 (2021): 138133).
5. Page 2, 'Mobility and self-diffusion peculiarities of water and Li+, Na+, Cs+ cations in Nafion membrane with well known ionogenic channel nanostructure and in MSC membrane revealed by 1H, 7Li, 23Na, 133Cs spin-relaxation and PFG NMR techniques are discussed.' If in the manuscript, only Nafion and MSC membranes are investigated, the title should not be 'Sulfonic Cation Exchange Membranes', since not many CEMs with sulfonic groups are studied yet.
6. Page 3, since N117 is used and the thickness is already known, why is the thickness of MSC membranes not mentioned? It should be noted that even Nafion membranes with different series demonstrate various physical-chemical properties, as well as different performances (International Journal of Energy Research 43.14 (2019): 8739-8752).
7. Page 10, Figure 4, 'Figure 4 [6]. Structure of the amorphous part of a perfluorinated sulfonate cation-exchange membrane. (1) Polymer backbone; (2) hydrated counter-ions and functional groups at a low moisture content; (3) transport channels for ions and water molecules at a high moisture content.' It should be explained how the proton/cation conductions are formed related to the phase separation between the hydrophobic and hydrophilic domains inside the CEM when hydrated (Ionics 25 (2019): 4219-4229). And it should be clear that the polymer backbone is typically hydrophobic, and sulfonic acid groups are hydrophilic.
8. Page 13, 'It has been shown that the limiting stage of macroscopic cation transfer ion transport is ion transport in the regions with a lower concentration of sulfur groups.' The word ''ion transport' should not appear twice.
For grammar issues, it is suggested that the author double-check the small grammar errors in the full text, especially the lack of and redundant use of definite articles.
Author Response
Dear Reviewer 1, authors thank for your the deep and valuable remarks. We are sure that remark corrections have essentially improved the manuscript.

Reviewer 2 Report
This manuscript deals with cation hydration and mobilities in different spatial scales in cation-exchange membranes. Experimental results obtained from the spin relaxation and pulsed field gradient NMR of 1H, 7Li, 23Na, 133Cs nuclei are quite interesting. However, I think that several points in the manuscript are unclear. I cannot judge the validity of the authors’ approach using simplified ionic transport model and cannot follow the discussion particularly on unique transport behavior of Cs+. The present manuscript should be improved before acceptance. My concerns are below.
1. The authors discussed the effect of cation species on the transport properties through cation exchange membrane using the simplified approach. The counter ion species of the polyelectrolytes intrinsically influence the aggregation structures of the solid-state polyelectrolytes or ion-exchangers (e.g., ionic domain size, connectivity between ionic domains, and perhaps local dielectric constant near –SO3 group) in particular for Nafion. Should such effects not be taken into account for the discussion on the ionic transport through the membranes? The authors should make this point clear in the main text.
2. Concerning the macroscopic self-diffusion coefficients measured by pulsed field gradient NMR. The diffusion lengths of the all ionic species under PFG NMR measurement conditions should be included in the main text.
3. Figure 1: Why are the measurement temperature different between samples? The reason should be included in the main text.
4. Line 201: “Water self-diffusion coefficient may be estimated from Einstein relation.”
Is valid Equation (10) in the solid polyelectrolyte such as nano-scaled microphase-separeted structure? The evidence for the validity should be included in the main text.
5. Table 2: The measurement temperature should be included.
6. Table 3: Why are there two data sets for -30°C?
Minor
1. The terms "alcaline" and "alkaline" are used in the title and main text, respectively. These terms should be unified.
2. What is the abbreviation of “MSC”? Polyethylene with grafted sulfonated polystyrene? Sulfocation-exchange? The definition of “MSC” should be included in Abstract and in the main text (for the first appearance in the Line 61).
3. Line 106: What is the role of P2O5?
4. Table 1: What is “MCK”? For the lambda s values, please correct “comma” to “dot”.
5. Line 249: T is the absolute temperature.
I think the manuscript could be more concise.
Please carefully review the English text before resubmitting.
Author Response
Dear Reviewer 2, authors thank for your the deep and valuable remarks. We are sure that remark corrections have essentially improved the manuscript.

Round 2
Reviewer 2 Report
I have carefully read the authors’ responses and the revised manuscript. The authors have adequately responded to the reviewer’s comments.